# Accurate Cardiac Duration Detection for Remote Blood Pressure Estimation Using mm-Wave Doppler Radar [note 1]

**DOI:** 10.3390/s25030619

**Published:** 2025-01-21

**Authors:** Shengze Wang, Mondher Bouazizi, Siyuan Yang, Tomoaki Ohtsuki

**Affiliations:** 1Graduate School of Science and Technology, Keio University, Yokohama 223-8522, Japan; wang@ohtsuki.ics.keio.ac.jp; 2Department of Information and Computer Science, Keio University, Yokohama 223-8522, Japan; bouazizi@ohtsuki.ics.keio.ac.jp (M.B.); yang@ohtsuki.ics.keio.ac.jp (S.Y.)

**Keywords:** blood pressure, Doppler radar, cardiac movement detection

## Abstract

This study introduces a radar-based model for estimating blood pressure (BP) in a touch-free manner. The model accurately detects cardiac activity, allowing for contactless and continuous BP monitoring. Cardiac motions are considered crucial components for estimating blood pressure. Unfortunately, because these movements are extremely subtle and can be readily obscured by breathing and background noise, accurately detecting these motions with a radar system remains challenging. Our approach to radar-based blood pressure monitoring in this research primarily focuses on cardiac feature extraction. Initially, an integrated-spectrum waveform is implemented. The method is derived from the short-time Fourier transform (STFT) and has the ability to capture and maintain minute cardiac activities. The integrated spectrum concentrates on energy changes brought about by short and high-frequency vibrations, in contrast to the pulse-wave signals used in previous works. Hence, the interference caused by respiration, random noise, and heart contractile activity can be effectively eliminated. Additionally, we present two approaches for estimating cardiac characteristics. These methods involve the application of a hidden semi-Markov model (HSMM) and a U-net model to extract features from the integrated spectrum. In our approach, the accuracy of extracted cardiac features is highlighted by the notable decreases in the root mean square error (RMSE) for the estimated interbeat intervals (IBIs), systolic time, and diastolic time, which were reduced by 87.5%, 88.7%, and 73.1%. We reached a comparable prediction accuracy even while our subject was breathing normally, despite previous studies requiring the subject to hold their breath. The diastolic BP (DBP) error of our model is 3.98±5.81 mmHg (mean absolute difference ± standard deviation), and the systolic BP (SBP) error is 6.52±7.51 mmHg.

## 1. Introduction

Globally, cardiovascular diseases (CVDs) are currently among the leading causes of mortality and stroke [1]. Regular blood pressure monitoring is crucial for the early detection and diagnosis of CVDs [2,3,4]. Nevertheless, conventional methods for monitoring blood pressure (BP) typically involve the use of a mercury sphygmomanometer and a stethoscope cuff, which might potentially cause inconvenience for the patient. Furthermore, it should be noted that the cuff-based technique lacks the capability to monitor BP continuously. In order to address this issue, researchers have been exploring the feasibility of calculating blood pressure (BP) using photoplethysmography (PPG) and electrocardiogram (ECG) signals [5,6,7]. While certain studies achieved acceptable precision, these techniques still need the placement of multiple electrodes and sensors on different regions of the subject’s body, resulting in patient discomfort and inconvenience [8,9,10].

Consequently, scientists have been researching the practicality of performing blood pressure measurements without physical contact [11,12,13,14,15,16,17,18]. They were inspired by the fact that PPG and ECG signals are all related to cardiac and pulse activities. In recent years, there has been considerable progress in the field of biomedical radar, resulting in its enhanced ability to detect subtle body motions. Non-contact blood pressure measurement appears to carry some potential. Specifically, the heart motion induces slight chest displacement, which can be detected with Doppler radar.

The researchers claimed that there exists a correlation between the heart activity captured by a radar system and BP measurements [11,12,19]. However, the identification of chest displacement resulting from heart motions poses a major obstacle. Due to the insignificance of these displacements in comparison with respiration and other body movements, the accuracy of recorded cardiac motions may be susceptible to contamination.

In some previous works, the participants were asked to hold their breath and remain still while the recordings were being made. Consequently, the influence of respiration is eliminated, and the signal-to-noise ratio (SNR) of the pulse wave is preserved at a high level. Most previous works obtain the pulse-wave signal by applying a band-pass filter (BPF) within a certain frequency range (i.e., [0.75∼2.00] Hz). The output of this BPF was assumed to be the pulse wave signal. This pulse wave signal was further used to extract cardiac features. Nevertheless, certain arbitrary bodily movements and noises continue to persist within the specified frequency range, and the pulse wave signal can only approximate the real cardiac action.

This work presents two novel approaches for extracting cardiac timing features, which can subsequently be utilized for non-contact blood pressure estimation. We prioritize deriving relatively precise cardiac features from raw radar signals, and the extracted precise features can lead to improved accuracy in estimating blood pressure. In our previous work [20], we employed the hidden semi-Markov model (HSMM) to estimate cardiac timing, achieving accurate blood pressure estimations. In this study, we introduce the U-net model for cardiac timing estimation. This new approach surpasses HSMM as it incorporates an innovative loss function, enhancing the overall accuracy of our predictions. The accuracy of the cardiac features is evaluated through a comparison with the gold standard electrocardiogram (ECG) signals, and the resulting estimation of BP is also presented. The experimental results indicate that the features obtained through our suggested methodology closely resemble those of the electrocardiogram (ECG) signals. Furthermore, the features produced by our suggested method obtain greater prediction accuracy when the same features are applied to BP prediction. Our contributions can be summarized as follows:Our research concentrates on high-frequency vibrations of the chest associated with cardiac activity. These high-frequency vibrations experience relatively less noise interference, allowing for a more accurate capture of cardiac movements.We utilize the hidden semi-Markov model (HSMM) to estimate systolic and diastolic times, taking advantage of probability changes in high-frequency signals.We enhance HSMM’s performance by integrating a new loss function within the U-net model, thereby improving the accuracy of cardiac timing feature detection.

The structure of this paper is as follows: Section 2 summarizes and discusses related work. Subsequently, the preliminary work is presented in Section 3. The proposed method is described in Section 4. In Section 5, we evaluate the performance of our proposed method and discuss its strengths and limitations. Finally, we conclude this paper in Section 6.

## 2. Related Work

This section presents the latest developments in BP estimation using radar-based methods. Presently, the majority of studies have employed the calibration-based approach.

Many studies have concluded that the pulse transit time (PTT) has a significant impact on BP values. Buxi et al. [13] used electrocardiogram (ECG) and radar signals to derive PTT and pulse arrival time (PAT). A linear regression model was subsequently employed to achieve calibration between PTT, PAT, systolic blood pressure (SBP), and diastolic blood pressure (DBP). Kuwahara et al. [21] computed the pulse wave velocity (PWV) using PTT and assessed the associations between PWV and BP. Nevertheless, the utilization of several sensor attachments remains difficult and inappropriate for the purpose of long-term blood pressure monitoring. Heng et al. [12] introduced a PTT extraction technique that utilizes simply one Doppler radar to mitigate the inconvenience caused by sensor attachments. In their method, the raw signal was first processed by BPF to generate the pulse wave, from which PTT was extracted. Via calibrating methods, they also discovered a linear relationship between PTT and BP.

The significance of time duration factors in addition to PTT has been mentioned in other previous works. Ohata et al. [14] obtained and exploited the systolic and diastolic times of each heartbeat. Strong correlations were observed between the two cardiac timings and BP. Kawasaki et al. [15] similarly observed a linear correlation between cardiac timing and blood pressure values within the same participant. Moreover, they utilized the correlation to estimate the fluctuations in BP for the study subject. While the aforementioned approaches demonstrate the ability to estimate blood pressure, it is important to note that each of these methods demands a calibration stage for each test subject. However, the linear relationship between a single feature and blood pressure values varies from subject to subject, leading to models that have limited generalization ability.

To address this problem, researchers explored machine learning methods to estimate blood pressure without relying on a calibration process. In a work proposed by Jung et al. [16], multiple features were derived from the radar signal. Other than the time domain, feature engineering is also applied to the frequency domain as well. These features are later used as inputs to a support vector machine for BP value estimations. Nevertheless, the evaluation conducted in their work has solely focused on the mean related error (MRE), and the precise accuracy of blood pressure value remains uncertain. Shi et al. [17] manually chose six features from the pulse wave signal, and assessed the importance of every feature by calculating the correlation coefficient for each feature and the BP values. The three most important features were further used as input to a random forest model for BP value estimation.

The performance was evaluated by mean absolute error (MAE) and standard deviation (STD). Zheng et al. [18] introduced a blood pressure estimate technique utilizing dual radar. The wrist and chest pulses are subjected to feature engineering. The chosen features are then fed to an artificial neural network (ANN) to enable calibration-free blood pressure prediction. Radar-based systems have gained attention for their ability to measure physiological parameters non-invasively. Vysotskaya et al. [22] demonstrated the feasibility of using 60 GHz radar to monitor blood pressure by detecting the mechanical movements of the skin due to arterial pulsations. However, their study highlights the challenges of achieving clinical accuracy, suggesting the need for the further optimization of sensor technology and signal processing algorithms. Shi et al. [23] proposed to evaluate signal quality indices for radar-based blood pressure monitoring. Unlike traditional methods that may discard entire datasets due to poor signal quality, their technique evaluates each pulse wave individually, allowing for the retention of valuable data by excluding only those pulses that fail to meet quality standards. This method filters out low-quality pulse data without requiring a complete dataset rejection. Ye Qiu et al. [24] utilized a stacked deformable convolution network (RSD-Net) to enhance feature extraction from radar signals. This approach combines Kalman filtering, multiscale band-pass filters, and a periodic extraction method to pre-process the signals, capturing fine details of chest micro-variations indicative of cardiac motions, achieving Pearson correlation coefficients (PCCs) of 0.838 for systolic and 0.797 for diastolic blood pressure.

According to previously stated works, Ohata et al. [14] established a mathematical model describing the relationship between BP values and cardiac time features (i.e., systolic diastolic duration). Similarly, Shi et al. [17], Jung et al. [16], and Kawasaki et al. [15] also included cardiac timing features for calculating BP values IBI (i.e., systolic diastolic duration, IBI). Nevertheless, the actual cardiac activities can only be approximated by their feature extraction techniques.

The systolic time refers to the period during which the heart contracts and releases blood, while the diastolic time describes the duration during which the heart relaxes following the contraction. Multiple previous works like [14] applied a BPF in the frequency range of [0.75∼2.00] Hz to obtain the pulse wave signal from raw radar data. These approaches were established based on the hypothesis that the phase signal of the chest area reduces during cardiac contraction, and then assumed this period as the systolic duration. In the same way, it was presumed that the heart undergoes expansion as the phase signal increases, triggering the capture of the diastolic time. Figure 1a shows an example of the feature extraction method used in previous works. However, the systolic phase of the heart actually begins with the contraction of the ventricles and ends with the closure of the aortic and pulmonary valves [25]. Both activities result in the generation of brief, high-frequency vibrations inside the chest cavity. Furthermore, it should be noted that systole is observed immediately following the R-peaks of the ECG, whereas diastole is synchronized with the end of the T-wave in the ECG signal [26].

Overall, multiple works have acknowledged the significance of both diastolic and systolic times in BP estimation tasks. The timing durations obtained from previous studies may still exhibit noise and temporal delays. Furthermore, they can only approximate the true cardiac features. This paper is an extended version of our previous work [20], which has introduced machine learning techniques for a more accurate cardiac timing estimation. In this study, we leveraged the robust segmentation capability of the U-net structure to enhance the estimation of cardiac timing features. Utilizing these improved features, we subsequently achieved better performance in blood pressure estimation.

## 3. Preliminaries

### Doppler Sensor Functioning

In Figure 2, we show the system model and the setup of the Doppler sensor for capturing the heartbeat signal. The Doppler sensor, as suggested by its name, makes use of the Doppler effect to detect phase and frequency changes. In such a system, a transmitter Tx emits a microwave signal T(t) as follows:(1)$$T\left(t\right)=\cos\left(2\pi ft+\Phi (t)\right),$$
where *f* and Φ(t) are the carrier frequency and the initial phase, respectively. The signal bounces on the moving object (i.e., the chest), and the reflected signal is captured by the receiving antenna Rx. The received phase signal R(t) is expressed as follows:(2)$$R\left(t\right)=\cos\left(2\pi ft-\frac{4\pi d_{0}}{\lambda }-\frac{4\pi x(t)}{\lambda }+\Phi \left(t-\frac{2d_{0}}{c}\right)\right),$$
where $d_{0}$ is the distance between the Doppler sensor and the moving object, λ is the wavelength of the carrier, and x(t) is the variation with reference to $d_{0}$. By down-converting the received signal R(t), it is possible to obtain the baseband signal as follows:(3)$$B\left(t\right)=\cos\left(\theta +\frac{4\pi x(t)}{\lambda }+\Delta \Phi \left(t\right)\right),$$
where θ is the default phase shift, which can be calculated using $d_{0}$ and *f*, and ΔΦ(t) is the total residual noise. The quadrature mixer shifts the phase of B(t) by π/2, allowing for acquiring the components I(t) and Q(t) with a phase difference of π/2, where I(t) and Q(t) denote the in-phase and quadrature components, respectively. The in-phase component I(t) corresponds to the real part of the complex signal and is in phase with the transmitted radar signal. The quadrature component Q(t) represents the imaginary part and is orthogonal to I(t). These components allow for the extraction of amplitude and phase information from the radar signal. By analyzing the variations and interactions of I(t) and Q(t), we can obtain the dynamics of the observed targets, such as speed, distance, and angular position. I(t) and Q(t) are expressed as follows:(4)$$I\left(t\right)=\cos\left(\theta +\frac{4\pi x(t)}{\lambda }+\Delta \Phi \left(t\right)\right),$$(5)$$Q\left(t\right)=\cos\left(\theta -\frac{4\pi x(t)}{\lambda }+\Delta \Phi \left(t\right)\right).$$

As can be seen in the system given in Figure 2, a band-pass filter (BPF) is used to roughly remove undesired frequencies that do not include movement-related information. An operational amplifier (OP-AMP) is finally used to amplify the signals I(t) and Q(t).

In the current work, we use the I(t) and Q(t) signals individually, and further combine them into a complex signal S(t) defined as follows:(6)S(t)=I(t)+j·Q(t),
where *j* is the imaginary number satisfying $j^{2}=-1$. S(t) can therefore be expressed also as(7)$$S\left(t\right)=e^{j\left(\pm \frac{4\pi vt}{\lambda }+\Phi \left(t\right)\right)},$$
where *v* is the speed of the target, and ± indicates the direction of the motion. The frequency shift value $f_{shift}$ induced by the object movement is(8)$$f_{shift}=\frac{4\pi vt}{\lambda }\;\cdot \;\frac{1}{2\pi t}=\frac{2v}{\lambda }.$$

## 4. Proposed Method

This section presents a calibration-free blood pressure measurement method utilizing Doppler radar, which operates while the subject maintains normal respiration. Under such conditions, the cardiac-induced displacements in the thoracic surface are comparatively insignificant when compared with those induced by respiration or bodily movements. Consequently, pulse wave signals are susceptible to contamination from breathing and surrounding noise.

We focus on the higher-frequency vibrations generated by the cardiac muscle activities in the thoracic region, as opposed to the pulse wave utilized in previous research. Specifically, as shown in Figure 3, each heartbeat is introduced by systole and diastole behaviors.

The heart’s ventricle contraction triggers the systole activity, and it ends with the sealing of the aortic and pulmonary valves [25]. The above-mentioned activities introduce vibrations in the chest area. Moreover, the extraction of cardiac features is conducted based on the high-frequency vibrations. We propose to use the integrated spectrum as input to depict the vibration signals for feature engineering. The energy changes in the subject’s chest area during high-frequency vibration can be more accurately captured by the integrated spectrum. Two machine learning algorithms are proposed for cardiac timing estimation using the integrated spectrum as input. The results demonstrated that our proposed methodology can improve the accuracy significantly in predicting cardiac duration features. Compared with the features extracted by pulse wave signals, our features could achieve higher BP estimation accuracy. A flowchart of our proposed method is shown in Figure 4. Here, the proposed method consists of three steps: (i) pre-processing and spectrum integration, (ii) cardiac feature estimation, and (iii) BP estimation.

### 4.1. Pre-Processing and Spectrum Integration

#### 4.1.1. Band-Pass Filter

The first step is applying a BPF. As previously established, the frequency range of [0.5∼3.0] Hz contains the heart rate information and generated pulse wave signal. Previous works [14,15,16] have all applied feature engineering to the pulse wave signal obtained through roughly such frequency bands. However, the onset of systolic activity is marked by the contraction of the heart’s ventricles and ends with the closure of the aortic and pulmonary valves [25]. This process generates brief, high-frequency vibrations in the chest region. That being the case, our work concentrates on the high-frequency oscillations in the chest area, which are triggered by the heart muscle movements. Here, a BPF with a higher frequency band [8.0∼30.0] Hz was applied, and the results contain important information related to cardiovascular activities, specifically systolic and diastolic time duration.

In addition, this range is carefully chosen as it predominantly includes the frequencies at which cardiovascular movements occur. Cardiovascular movements, unlike large body movements or respiratory functions, exhibit periodic behavior that is more predictable and isolated within this higher-frequency band. Moreover, this frequency range also excludes the high-frequency noise components in the radar hardware, which ensures that the data acquisition process is less susceptible to the random noise typically associated with electronic devices and external environmental factors. By filtering our signal to capture only these frequencies, we significantly reduce the interference from non-cardiovascular sources, enhancing the reliability of our measurements.

#### 4.1.2. Short-Time Fourier Transform

A short-time Fourier transform (STFT) is then applied to the filtered signal output from the previous step. The STFT has several parameters that need to be tuned to capture the information needed to obtain the heart muscle movements that allow for accurately obtaining the cardiac activities. The main parameters of the STFT are as follows:The length of the segment (window) over which the Fourier transform is calculated, referred to as $W_{stft}$.The step size indicates the distance between the starting points of two consecutive windows. This can be inferred otherwise from the amount of overlapping between the segments. The step size is referred to as $S_{stft}$.

For the window length in STFT, it is advisable to use a relatively short window length and step size as they can provide better temporal resolution. The time window should also be shorter than the interbeat interval (IBI) to ensure that each window captures only one heartbeat. In our approach, the window length $W_{stft}$ is set to 256 ms, and the step size $S_{stft}$ is 1 ms.

Initially, we compare the spectrogram of the traditional pulse wave signal [27] with our signal that has been filtered for higher frequencies. As shown in Figure 5, the spectrogram of the traditional pulse wave contains noise beyond cardiac activities, while our filtered signal exhibits distinct energy scattering that aligns with the cardiac cycle. Systole and diastole are two essential phases of the cardiac cycle. Identifying when these phases occur can yield accurate cardiac motion. We employ an integrated spectrum waveform derived from STFT for this purpose. The integrated spectrum is calculated by summing the energy across the frequency spectrum for each step within the range of [5∼30 Hz], as shown in Figure 6. Referring to Figure 6, the cardiac movement in the pulse wave signal appears distorted as it fails to align with the ground-truth ECG. In contrast, our integrated spectrum reveals two distinct peaks in each heartbeat. When aligned with synchronous ground-truth ECG signals, the first peak in the integrated spectrum emerges immediately after the R-peaks, while the second peak appears at the conclusion of the T-wave. These occurrences coincide with the phases of systole and diastole. From this, we can deduce that the initial peak may be caused by the synchronized contraction of the heart muscle before systole, and the latter peak may result from the vibrations due to the closing of the valves, marking the start of diastole. Further analysis can be conducted by extracting features based on the systolic and diastolic durations.

In this work, the integrated spectrum is first segmented, defining that the duration is set to 10 s each. Feature extraction is applied on each segment, utilizing the average values of the features and BP for each segment to create the training data. The preparation of ECG signals involves a necessary pre-processing step. To derive the ground-truth ECG, the raw signal is subjected to a band-pass filter that operates within the [0.24∼31.25 Hz] frequency range.

#### 4.1.3. Spectrogram Integration

Integrating the spectrogram refers to the operation of summing the energy over a certain range of frequencies, resulting in a time-domain signal. The integrated signal is rich of heart muscle movement features. As it stands, this signal contains enough information to reconstruct the ECG signal, though it might contain noise, making the detection of R-peaks not always accurate. A copy of this output signal is kept for the evaluation of the ECG reconstruction.

### 4.2. Cardiac Feature Estimation

Drawing from the previously discussed details, the integrated spectrum offers the capability to precisely identify the timing of systole and diastole. This work proposes a logistic regression-based hidden semi-Markov model (HSMM), inspired by the methodology in [26], to determine the timings of cardiac activities such as systolic time, diastolic time, and the interbeat interval (IBI) [28]. Furthermore, a deep learning algorithm for signal segmentation has also been employed to estimate features relevant to cardiac timing.

#### 4.2.1. Hidden Semi-Markov Model

Based on the previously stated information, the integrated spectrum offers the potential to accurately determine the timings of systole and diastole. In this regard, we propose a logistic regression-based hidden semi-Markov model (HSMM), inspired by [26], to estimate the timings of cardiac activities, specifically the systolic time, diastolic time, and IBI. Here, our HSMM model has similar concepts to a state duration dependency hidden Markov model, as proposed in [28]. In particular, the matrix *A* in the traditional Markov model represents the transition matrix. In our scenario, the matrix *A* exhibits a “nonergodic” characteristic, as each state can only be reached from one specific preceding state. This means that the progression of states follows a predetermined sequence. The probability of transitioning from one state to another is solely dependent on the duration for which the current state has persisted and on the observations from the integrated spectrum. The probability of a state transition increases with the length of time the system remains in its current state.

In our study, we begin by discussing the conventional hidden Markov model (HMM), which serves as the groundwork for our more complex model adaptations. The original HMM framework is defined byλ=(A,B,π).

In this model, *A* serves as the matrix that depicts the transitions between states, *B* describes the probability distribution of observations, and π specifies the distribution of states at the start. The sequence of observations is represented by $O=O_{1},O_{2},\cdots ,O_{T}$, with $O_{t}$ being the specific observation vector at the time step *t*.$$A=\left\{a_{ij}\right\}.$$

The matrix *A* determines the probability of transitioning between states at each time step. For example, if the current state is *i* and the current time step is *t*, state transitions occur when the system moves from state *i* to the next state *j* at the following time step t+1.$$B=\left\{b_{j}\left(O_{t}\right)\right\}.$$

The matrix *B* specifies the probability of observing $O_{t}$ given that the system is in state *j*. Define $\delta_{t}\left(j\right)$ as the most likely sequence of states for the initial *t* observations, and it ends in state *j* at time *t*. The initial value $\delta_{1}\left(j\right)$ can be determined as $\pi_{j}b_{j}\left(O_{1}\right)$. By taking into account the information from previous time steps, $\delta_{t}\left(j\right)$ for the conventional HMM can be iteratively calculated by$$\delta_{t}\left(j\right)=\left[\max_{\left(1\leq i\leq N\right)}\delta_{t-1}\left(i\right)a_{ij}\right]\;\cdot \;b_{j}\left(O_{t}\right).$$

Transitioning from this traditional approach, our model introduces modifications that transform the standard HMM into a hidden Semi-Markov model (HSMM). In the HSMM, the transition probabilities and state durations are governed by a modified transition matrix *A* and a duration-dependent parameter *p*, enhancing the model’s ability to account for time-based characteristics specific to cardiac activities. Specifically, the matrix *A* in our HSMM is adapted to reflect nonergodic properties, and an additional parameter *p* is required to describe the relationship between the duration of time spent in a state and the likelihood of transitioning to another state. By incorporating the probability density of duration, $\delta_{t}\left(j\right)$ for the HSMM can be defined as follows [28]:$$\delta_{t}\left(j\right)=\max_{d}\left[\max_{i\neq j}\left[\delta_{t-d}\left(i\right)\;\cdot \;a_{ij}\right]\;\cdot \;p_{j}\left(d\right)\;\cdot \;\prod_{s=0}^{d-1}b_{j}\left(O_{t-s}\right)\right].$$

Here, $p=\left\{p_{i}\left(d\right)\right\}$ denotes the probability of staying in the same state *i* for a duration equal to *d*. Figure 7 presents an example of our HSMM algorithm. In this approach, a four-state HSMM is employed to estimate the duration of various activities within a single heartbeat.

Our HSMM model is inspired by similar approaches [26] that employ contact devices to segment the heart sound. In these methods, a typical set of states is usually used. They include (1) the initial chest vibration, (2) the systolic time duration (i.e., the duration separating the second chest vibration from the first one), (3) the second chest vibration, and (4) the diastolic time duration (i.e., the duration separating the next initial chest vibration from the current second one). Furthermore, these states follow a predetermined sequence, with each state being accessible only from a specific preceding state. For instance, the systolic period can only be entered following the first chest vibration and is not accessible from any other state.

In this study, the HSMM is solely focused on predicting the time steps at which transitions between states occur. Given the fixed order of state changes, a sequence of state transitions can be generated from an initial state using the predicted time steps. The resulting sequence of state changes, and the corresponding integrated spectrum is depicted in Figure 8. As HSMM goes, we first calculate the state transition probability at each time step. The Viterbi algorithm is then employed and takes as input these calculated probabilities to determine the most likely sequence of state changes (i.e., one that yields the highest overall probability). The calculation of the probability of a state transition at each time step takes into account both the observations from our integrated spectrum and the duration of the current state.

#### 4.2.2. U-Net

Building upon the foundational knowledge that the HSMM serves as a machine learning algorithm specifically tailored for signal segmentation, we introduce a modified U-net architecture. U-net is a deep learning algorithm composed of convolutional neural networks and was originally designed for tasks in image segmentation. This modified U-net architecture is tailored to surpass the HSMM in estimation accuracy, and incorporates enhancements suited for the complexities and specific requirements of signal segmentation.

U-net is a convolutional neural network architecture that was originally developed for biomedical image segmentation tasks. The architecture is characterized by its unique ‘U’-shaped structure, which comprises two main parts: a contracting (downsampling) path and an expansive (upsampling) path, as shown in Figure 9.

Contracting path: This part of the network follows the typical architecture of a convolutional neural network. It consists of a repeated application of convolutions, each followed by a ReLU and a max pooling operation for downsampling. With each downsampling step, the network increases the number of feature channels.

Expansive path: The second half of the network involves upsampling of the feature map, followed by convolutional operations. This part of the network also includes “skip connections” from the contracting path. These connections provide the expansive path with context information from the contracting path, which is crucial for the precise localization in image segmentation tasks.

The purpose of the U-net architecture is to provide a precise and efficient way of performing image segmentation, particularly useful in medical imaging where the accurate delineation of boundaries is crucial. The effectiveness of U-net comes from its ability to capture the context from the entire image while maintaining the ability to focus on fine details in specific areas of the image. We applied U-net for the following reasons:

Handling noise: One of the key challenges in processing integrated spectrum is the presence of various types of noise, such as muscle artifacts, baseline wander, and random noise in the high-frequency range. U-net’s multiple layers and depth help in learning to differentiate between noise and actual signal components. The network can learn to ignore or filter out irrelevant noise features through training, focusing instead on meaningful signal patterns.

Learning from variability: Our integrated spectrum can vary significantly between individuals due to differences in heart size, position, and health conditions. U-net’s capacity to learn from a wide range of examples allows it to generalize well across different types of cardiac movement signals. Training helps the network develop a robust model that can accurately process signals even when they deviate from the “typical” patterns seen in training data.

Enhanced by skip connections: These connections not only assist in preserving important details but also help in reinforcing the learning against noise. By reintegrating early signal representations directly with deeper layers, the network maintains access to raw, less abstracted features that can provide checks against over-generalization or misinterpretation of noisy data.

In this work, the U-net structure is customized to perform the segmentation task on our 1D signal data. Specifically, several key modifications were implemented to optimize the model for integrated spectrogram segmentation as follows:Convolutional layers: The standard 2D convolutional layers (Conv2D) were replaced with 1D convolutional layers (Conv1D). This alteration enables the network to effectively process time-series data along the temporal dimension.Pooling and upsampling: Traditional 2D pooling and upsampling operations were replaced with 1D variants to preserve the integrity of the temporal information throughout the network’s processing pipeline.Output layer adjustments: The output layer was specifically tailored to produce a 1D output that accurately corresponds to the segmented regions of the ECG signal, distinguishing between systole and diastole periods effectively.

The ground-truth label is generated through the ECG signal; specifically, in every heartbeat, the value of systole time is set to 0, while the value of diastole time is set to 1, as shown in Figure 10. The loss function we applied is MAE (mean absolute error), and finally, the U-net structure is employed. The detailed structure of the network is displayed in Table 1. In training our models, we utilized the Adam optimizer with an initial learning rate of 0.001. The network was trained using a batch size of 256 over 100 epochs, with mean absolute error (MAE) as the loss function and mean squared error (MSE) as an additional monitoring metric.

### 4.3. BP Estimation

Prior research [14,29] has established a strong correlation between the IBI, systolic time, and diastolic time with BP. In this study, these features can be accurately extracted using the outputs from our HSMM algorithm and U-net, both demonstrating a relatively low error rate. With the HSMM algorithm, the durations of systole and diastole are discerned at the time step of state transition. Beyond these temporal features, the width of the pulse wave at a level that is 25% of its maximum peak is also incorporated as an additional feature.

A random forest model utilizes these extracted features to estimate SBP and DBP. Moreover, a grid search method is employed to determine the most effective combination of hyperparameters. The evaluation of performance across potential sets of hyperparameters is conducted, and the set yielding the highest accuracy is chosen.

## 5. Results and Discussion

### 5.1. Experimental Results

Within this section, we deploy our suggested methodology across several experiments to evaluate the significance of the features captured by our method. First, we provide a concise overview of the dataset employed in this study. Second, we assess the precision of the time durations associated with cardiac activities. Next, we evaluate the impact that the cardiac features, as estimated by our method, have on the task of blood pressure estimation. Last, we compare the accuracies of our blood pressure predictions with those from existing research to draw a clear conclusion.

#### 5.1.1. Dataset

To evaluate our BP estimation technique, we used the open-sourced medical dataset offered in [27]. In this dataset, the Doppler radar signals were recorded using two Doppler radars whose carrier frequencies are 24.25 GHz (New Japan Radio, NJR4262, Tokyo, Japan). Simultaneously, the ECG data (along with other vital signals) were recorded using a contact device attached to the subjects’ bodies. In addition, the synchronized non-invasive continuous BP data were also collected using a Task Force Monitor (TFM). A total of 25 subjects’ data were used in our experiments, 18 of whom were males and 7 were females. The participants had an average age of 30.7 ± 9.9 years and an average body mass index (BMI) of $23.2\pm 3.3\;{\mathrm{kg}/m}^{2}$. They were selected based on health screenings to ensure that they were free from any chronic illnesses and were not taking any medications known to influence cardiovascular functions. This healthy cohort was chosen to establish baseline cardiac timing and blood pressure estimations in a controlled environment, providing a foundation for further research in populations with varied health conditions. The subjects’ data were recorded while they were resting on a bed in a supine position. The data using the different devices were collected at a sampling frequency equal to 1 KHz, and were 16-bits-coded. For each subject, 10 min’s worth of data were recorded. The data were down-sampled to 125 Hz and split into segments of 10 s, with 5 s overlapping between consecutive segments. After removing recordings with low quality, we were left with 7069 usable data segments. We employed 80% of these for training purposes and used the remaining 20% for testing. Each generated sample was pre-processed to extract the following components, which we used later for evaluation:The subject ID: This is used later to divide our dataset during cross-validation to make sure that samples of the same subject do not leak from the training set to the validation set. In other words, samples from the same subject should be used exclusively in the training or the validation set.ECG signal: This is the raw ECG signal cleaned by applying a simple BPF with cutoff frequencies equal to 0.85 and 4 Hz.Raw I and Q signals: These are the raw Doppler I and Q signals with no filtering or pre-processing applied.Filtered I and Q signal: These are the Doppler I and Q signals filtered with a BPF whose cutoff frequencies are set to 0.5 and 2 Hz.Integrated spectrogram: The Doppler I and Q signals are first filtered with a BPF whose cutoff frequencies are set to 8.0 and 30.0 Hz, and spectrogram Integration is described in Section 4.1.2 and Section 4.1.3.Blood pressure value: The non-invasive continuous BP value recorded by the Task Force Monitor.

#### 5.1.2. Cardiac Timing Estimation Accuracy

Here, we assess the effectiveness of the HSMM and U-net models we have proposed. In this study, to evaluate the adaptability and robustness of our model across different physiological profiles, we employed a subject-specific five-fold cross-validation method. The dataset consisted of recordings from 25 subjects. In each fold, the data from 5 unique subjects were designated as the test set, while the data from the remaining 20 subjects formed the training set. This partitioning was strategically planned to ensure that each subject’s data were utilized as a test set in exactly one fold. The model’s performance was evaluated based on metrics calculated separately for each test set, and the overall performance was derived as an average of these results across all folds. This cross-validation scheme not only prevented data leakage but also ensured that the performance metrics reflected the model’s efficacy in handling variations across different subjects. The root mean square error (RMSE) and mean absolute error (MAE) for the estimated IBI, systolic time, and diastolic time were calculated, with the true values obtained from the ECG signal’s R-peaks and T-wave ends. A comparison of cardiac state transitions and the outputs from our U-net model against the actual ECG signal is presented in Figure 11. Additionally, we calculated RMSE and MAE for these cardiac features using the pulse wave data generated from BPF following the approach suggested by [14]. In Table 2, we present a comparison of our proposal with the conventional methods. As can be seen from the results presented in the table, our method performs much better than the conventional ones, reaching error values that are largely lower than those of the conventional ones. For instance, the MAE of our U-net-based method for the systolic and diastolic time durations reached 0.0049 s and 0.0059 s, respectively. For reference, these error values in the case of the BPF-based method, reached 0.1913 s and 0.1013 s, respectively, Similarly, the IBI MAE using the U-net reached only 0.0048 s, whereas it reached 0.1053 s for the BPF-based method. A similar pattern was observed for the RMSE as well. Our approach demonstrates precise detection and estimation of cardiac timing features, surpassing previous methodologies.

#### 5.1.3. Radar-Based BP Estimation

In our study, we utilized the random forest algorithm for estimating blood pressure (BP) values using the previously acquired cardiac features. We employed the grid search technique to tune the hyperparameters, iterating through multiple combinations of hyperparameter sets and selecting the one that yielded the lowest error rate. To demonstrate the usefulness of the features and to highlight the superiority of our feature estimation method in BP prediction, we compared the performances of these features when estimated using different techniques from the literature to ours. The performance was measured in terms of BP prediction error. Note that, while the features in all the methods were the same, the estimation of these features differed from one work to another. These features include systolic time, diastolic time, IBI, and the width of the pulse wave at 25% of its peak height, obtained using different methods: BPF, HSMM, U-net, and ECG.

Once the hyperparameters were chosen, the different methods’ generated sets of feature values were input to our random forest regressor to predict the subject’s BP. We evaluated the performance using mean absolute error (MAE) and standard deviation (STD). The results are presented in Table 3. As expected, the features derived from the actual ECG signal showed the greatest accuracy. In comparison, the features we precisely estimated surpassed those acquired through BPF, underscoring the importance of these cardiac timings in predicting BP. Additionally, the outcomes suggest that the cardiac timings produced by the HSMM and U-net methods are more precise, enabling our approach to attain a lower BP estimation error.

#### 5.1.4. Comparison with Previous Works

A comparison between our proposed method and some of the existing calibration-free methods proposed in the literature is given in Table 4. Our method has outperformed the results reported in [18]. When compared with [17], our method demonstrates competitive performance in terms of error rates (i.e., MAE and STD) of DBP and SBP, even in cases of normal breathing. This could be attributed to the fact that the features extracted in previous studies are not as accurate or representative. Earlier works used simple filtering techniques such as applying a BPF as a proxy for the participant’s pulse wave. However, the signals produced by these methods are only a rough approximation of actual cardiac activities and still contain noise and time lag. In contrast, our approach uses the integrated spectrum to eliminate the effects of respiration and random noise. Two machine learning algorithms result in more precise cardiac timings. By comparing our performance with that of other studies, it is evident that our method generates more accurate and representative features for BP estimation.

### 5.2. Discussion

To the best of our knowledge, no prior work used the high-frequency components of the Doppler signal to extract the systolic time and diastolic time the way we do. Not only is our work the first to incorporate them, but also the processing that comes along is part of the contribution of this paper.

In addition, it is important to acknowledge a few limitations. The dataset used in this study is exceptionally high quality, with minimal noise or interference. Although the subjects were breathing normally, they remained still and the background was free of distractions. The experimental setup included the use of radar technology, which, while highly sensitive to the subtle high-frequency movements associated with the heartbeat, requires precise positioning close to the subject’s chest to ensure accurate data capture. This proximity is necessary due to the radar’s susceptibility to external oscillating noises and its need to isolate minute cardiovascular movements from other noises. These methodological choices and the radar’s technical limitations were critical in shaping the study’s design and have informed the interpretation of our findings and the planning of future research directions. Real-world scenarios are significantly different, often involving varying distances, background interference, subject movement, and other noise factors. Consequently, the performance demonstrated in this study may not fully generalize to real-world data, where these challenges are more prevalent. Further validation on diverse and less controlled datasets is necessary to assess the robustness of our approach in practical applications. It is also important to note that the estimated BP values obtained through our approach are not intended to replace the standard mercury sphygmomanometer. Instead, the goal is to provide a non-contact BP measurement method capable of enabling long-term monitoring and facilitating the early detection of abnormalities. Finally, the BP estimation performance may decrease for unseen subjects. This is because, while systolic time and diastolic time have been shown to be effective features for BP estimation, the relationship between these time durations and BP values varies from person to person. This individual variability poses a challenge in achieving consistent accuracy across diverse subjects.

While the current study effectively utilizes a high-frequency band (8–30 Hz) to focus on cardiovascular movements and minimize interference from larger body movements and respiration, the challenge of patient movement during measurements remains a significant concern. To further enhance the accuracy and robustness of our radar-based blood pressure measurements in the presence of such movements, we have identified two promising techniques slated for exploration in our future research:

Ellipse Fitting for Enhanced Signal Quality: We plan to explore the application of ellipse fitting techniques in our signal processing framework. This method aims to more precisely model and extract periodic cardiovascular signals, even in the presence of background noise caused by patient movements. Ellipse fitting could provide a more accurate geometrical representation of the cardiovascular waveform, leading to better isolation and analysis of the relevant signals.

Application of Diffusion Models: Additionally, we intend to integrate diffusion models to refine the signal processing approach. These models are anticipated to effectively differentiate physiological signal changes from noise associated with movements, smoothing the radar data to focus more closely on genuine cardiovascular dynamics. This approach will be crucial for ensuring that our measurements reflect true blood pressure variations rather than artifacts introduced by extraneous movements.

By addressing these technical challenges through advanced signal processing techniques, our future studies will seek to improve the reliability and applicability of non-contact blood pressure monitoring systems in clinical and everyday settings. These enhancements will not only bolster the accuracy of our current model but also expand its utility across a broader range of operational scenarios.

## 6. Conclusions

In conclusion, this study presents a novel approach for blood pressure estimation that uses the integrated spectrum derived from the Doppler radar signal as input to a U-net and an HSMM algorithm to extract accurate cardiac timing features. Our method demonstrates superior performance in comparison with previous calibration-free works, particularly in terms of feature accuracy and representativeness. The features used in these works are extracted only with reference to the pulse wave and cannot, therefore, capture useful information related to the cardiac movements. This leads to a less accurate prediction of the BP. Our method, on the other hand, makes use of the integrated spectrum as input for two kinds of machine learning algorithms to estimate the cardiac features. The two algorithms used are the HSMM and a custom U-net that we made to account for the nature and size of the input data. The results indicate that precise extraction of cardiac timings, such as systolic time, diastolic time, and IBI, is crucial for reliable blood pressure estimation.

Our study focused on healthy individuals, predominantly aged 30 ± 10 years, which provides a controlled baseline for developing and validating our novel methodology for cardiac timing and blood pressure estimation. However, it is important to note that the physiological relationship between these parameters may differ significantly in populations with cardiovascular conditions due to factors like arterial stiffness and impaired baroreflex. Recognizing this, future applications of our method in clinical settings should consider the necessity for model recalibration or re-training tailored to specific patient profiles upon admittance. Such adaptations are essential to ensure that our methodology can be generalized effectively across varied patient demographics and conditions, thereby enhancing its clinical utility and accuracy.

In future work, we will try to explore the detection of additional cardiac features using the radar system, particularly those that are correlated with blood pressure values. While the focus on systolic time, diastolic time, and IBI has proven effective, there may be other physiological markers that can further enhance the accuracy and reliability of blood pressure estimation. Investigating a broader range of cardiac features could provide a more comprehensive understanding of cardiovascular health and improve the overall performance of non-contact blood pressure monitoring systems.

## Figures and Tables

**Figure 1 sensors-25-00619-f001:**
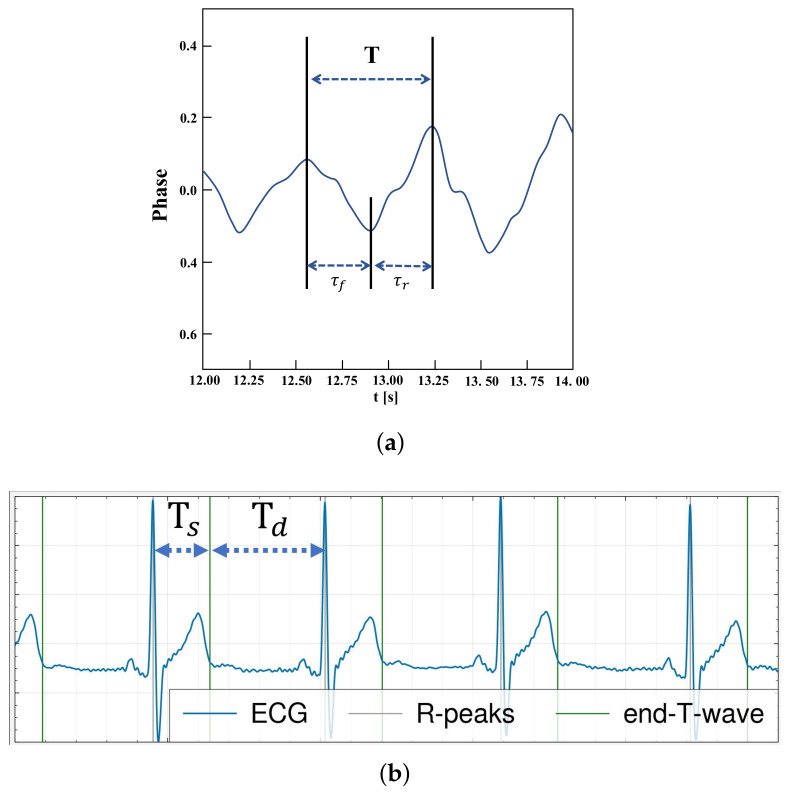
An illustration of (**a**) conventional assumption on systolic and diastolic timing extraction and (**b**) actual systolic and diastolic timings from ECG waveform.

**Figure 2 sensors-25-00619-f002:**
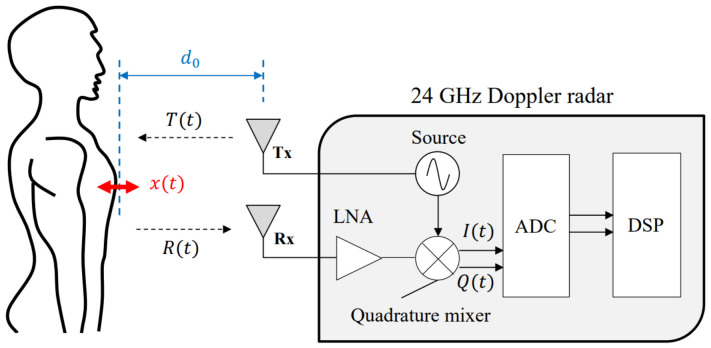
The system model and the setup of the Doppler sensor for capturing the heartbeat signal.

**Figure 3 sensors-25-00619-f003:**
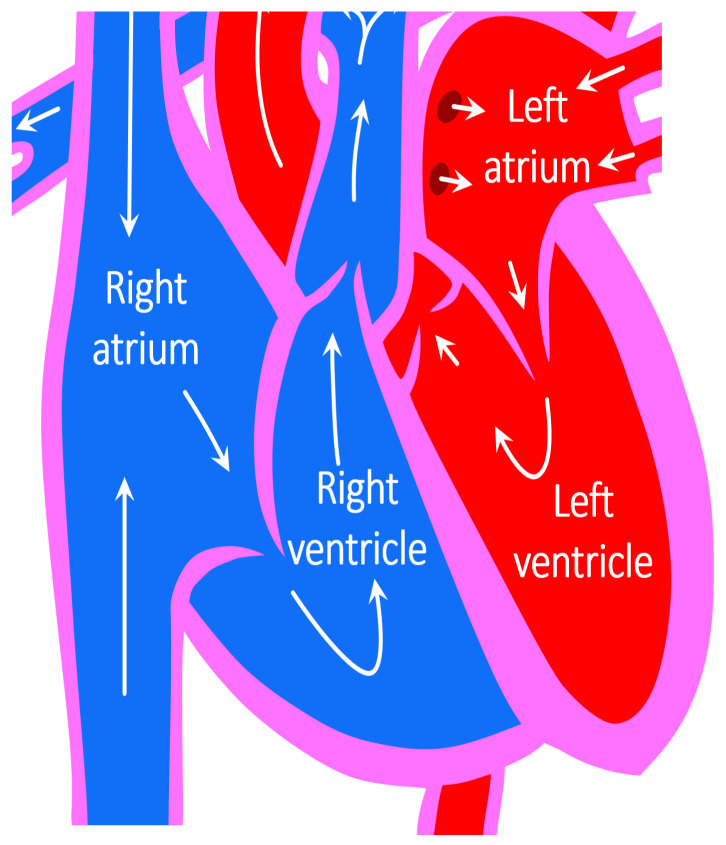
An illustration of the heart parts.

**Figure 4 sensors-25-00619-f004:**
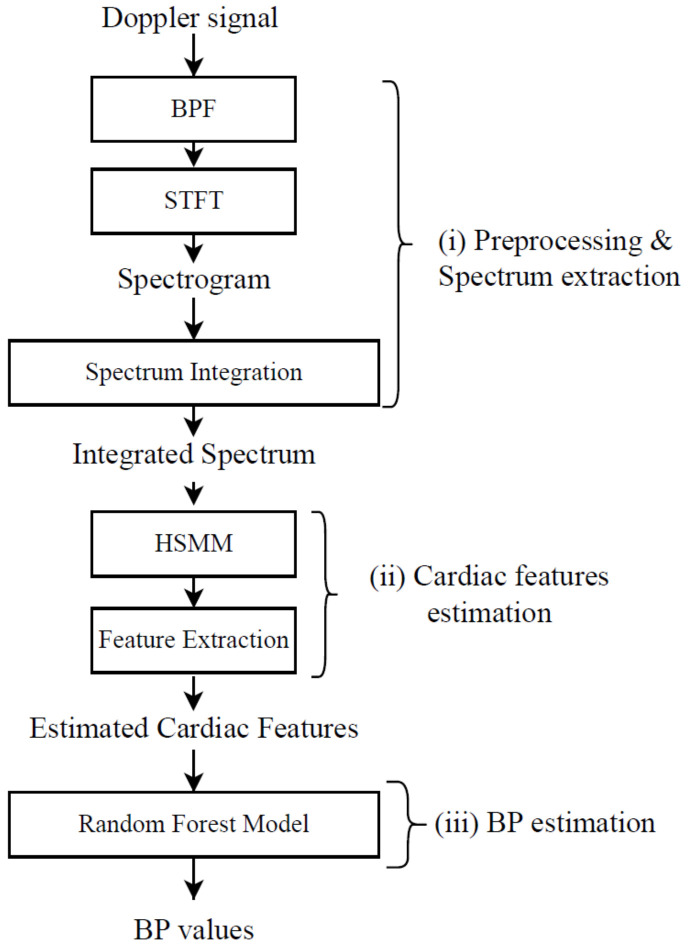
Flowchart of the proposed method.

**Figure 5 sensors-25-00619-f005:**
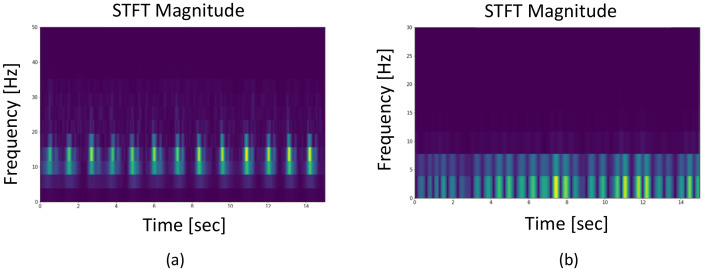
(**a**) The spectrogram of the conventional pulse wave signal; (**b**) the spectrogram of the higher-frequency radar signal selected in our work.

**Figure 6 sensors-25-00619-f006:**
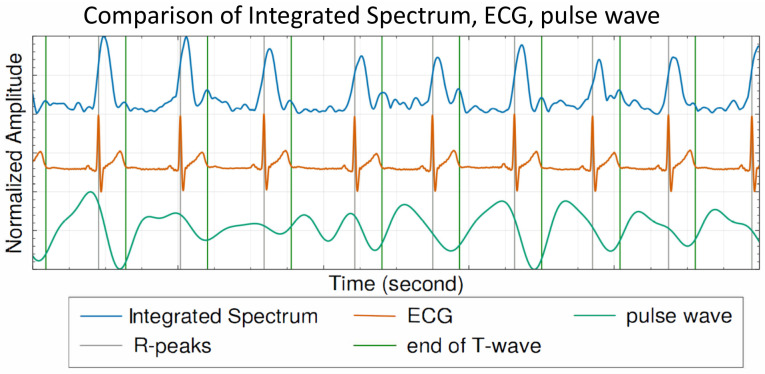
Integrated spectrum and pulse wave with ECG as gold standard.

**Figure 7 sensors-25-00619-f007:**
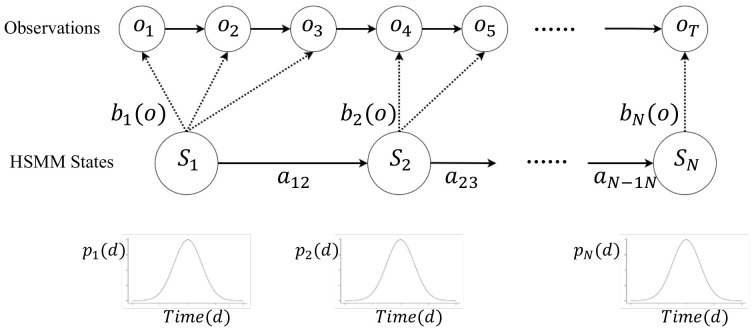
Example of an HSMM algorithm.

**Figure 8 sensors-25-00619-f008:**
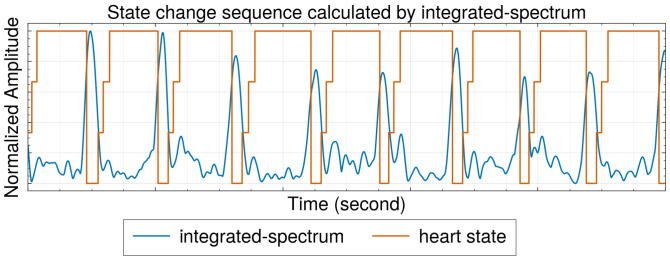
The estimated state change generated by HSMM.

**Figure 9 sensors-25-00619-f009:**
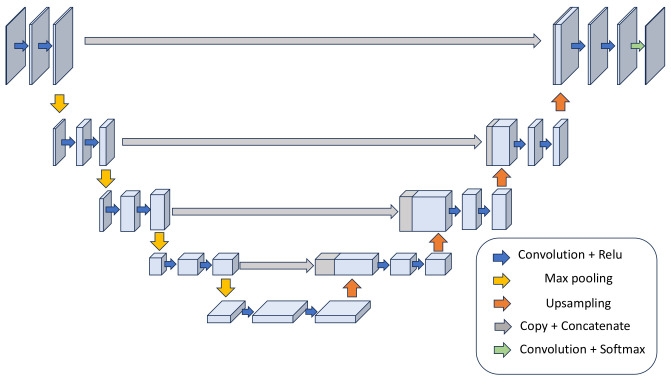
Example of a U-net structure.

**Figure 10 sensors-25-00619-f010:**
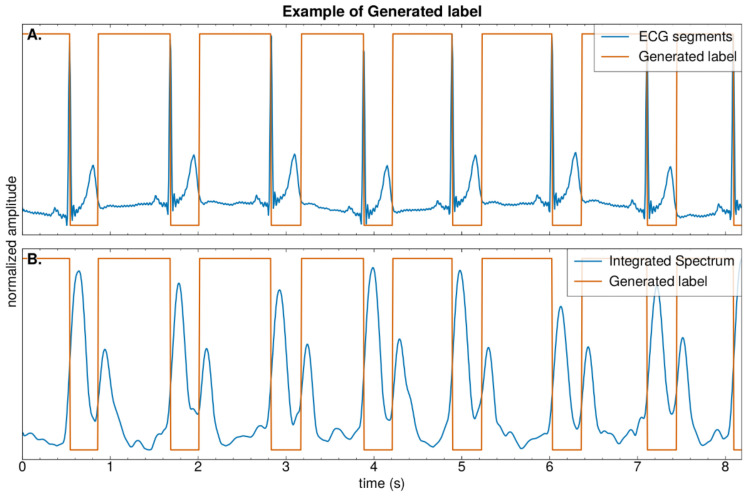
An illustration of (**A**) shows the label are generated from the ECG signal. (**B**) shows the comparison between Integrated spectrum and the generated label.

**Figure 11 sensors-25-00619-f011:**
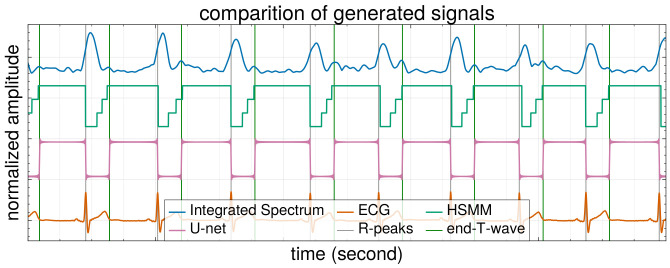
The integrated spectrum, state changes, and U-net results compared with the R-peaks and end of the T-wave in the ECG.

**Table 1 sensors-25-00619-t001:** Detailed U-net architecture.

Layer	Operation and Configuration	Output Size
Input Layer	Input: Single-channel input signal	(1024, 1)
Contracting Path		
Convolutional Blocks	Repeated operations for each block: 2 × Conv1D with increasing filters (32, 64, 128), size 3, ReLU, same padding, batch normalization Pooling: MaxPooling1D, pool size 2	(512, 32) (256, 64) (128, 128)
Bottom Block	2 × Conv1D with 256 filters, size 3, ReLU, same padding, batch normalization	(128, 256)
Expansive Path		
Up-sampling Blocks	Each block includes the following: Up-sampling: concatenate with output from corresponding contracting block 2 × Conv1D with reducing filters (128, 64, 32), size 3, ReLU, same padding, batch normalization	(256, 128) (512, 64) (1024, 32)
Final Convolutional Layer	Operation: Conv1D with 1 filter, size 1	(1024, 1)

**Table 2 sensors-25-00619-t002:** Cardiac duration evaluation.

	Error Compared with ECG	
	**MAE (s)**	**RMSE (s)**
Systolic (BPF)	0.1913	0.2218
Systolic (HSMM)	0.0381	0.0592
Systolic (U-net)	0.0049	0.0047
Diastolic (BPF)	0.1013	0.2783
Diastolic (HSMM)	0.0485	0.0820
Diastolic (U-net)	0.0059	0.0086
IBI (BPF)	0.1053	0.1384
IBI (HSMM)	0.0179	0.0616
IBI (U-net)	0.0048	0.0068

**Table 3 sensors-25-00619-t003:** The mean absolute error and standard deviation for different feature sets.

	MAE+STD (mmHg)	
	**DBP**	**SBP**
BPF	4.78±6.52	7.93±10.51
HSMM	4.27±5.84	6.63±8.95
U-net	3.98±5.81	6.52±7.51
ECG	3.95±5.36	6.46±7.23

**Table 4 sensors-25-00619-t004:** Comparison between the performance of the proposed method and those of previous works (mmHg).

	Dataset	Requirement	Method	STD (DBP)	MAE (DBP)	STD (SBP)	MAE (SBP)
Shi [17]	Closed source, 25 subjects	Hold breath	Filtering + Random Forest	3.85	6.30	6.73	7.20
Zheng [18]	Closed source, 27 subjects	Normal breathing	Filtering + ANN	8.12	6.69	12.13	9.51
Ours	Open source, 25 subjects	Normal breathing	HSMM + Random Forest	5.84	4.27	8.95	6.63
Ours	Open source, 25 subjects	Normal breathing	U-net + Random Forest	5.81	3.82	7.51	4.84

## Data Availability

The dataset used in this work can be obtained by contacting the authors of this work [27].

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
