# Peer review of "Accurate Cardiac Duration Detection for Remote Blood Pressure Estimation Using mm-Wave Doppler Radarâ€"

_sensors, 2025, doi:10.3390/s25030619_

Round 1
Reviewer 1 Report
Comments and Suggestions for Authors
Comment and observations
Dear authors, thank you for this very interesting paper. Accurate Cardiac Duration Detection for Remote Blood Pressure Estimation using mm-Wave Doppler Radar is very interesting the work.
This work presents a novel approach for blood pressure estimation that uses the integrated spectrum derived from the Doppler radar signal as input to a U-net and an HSMM algorithm to extract accurate cardiac timing features.
There are some recommendations to improve the quality of this contribution. Thank you for sharing this work and knowledge.
I recommend that the article be accepted but first it is necessary to correct it. Here are my comments:
1. It is necessary that all the manuscript must be revised and corrected because there are some grammatical errors.
2. Could you please provide more information about the patients who were the subjects of the study? Such as age, whether they have any illnesses, whether they take medication, etc., to focus the population with more elements. Include in the manuscript.
3. Page 2, line 75 the period is duplicated.
4. Define the term I(t) and Q(t).
5. It would be advisable to indicate all possible noises that may lead to an incorrect measurement and how they are considered to mitigate them. Include them in the article.
6. Page,7 lines 200 and 207 is missing the dash.
7. The quality of Figure 5 is low, improve the image.
8. Page 9, line 276 and 287, is missing the S in HMM.
9. From figure 10 correct the legend lable by label.
10. What modifications would be necessary if the patient moves too much when taking measurements? What changes to the model should be included?
Reviewer 2 Report
Comments and Suggestions for Authors
Q1: The details of the U-Net architecture are missing. It would be helpful if you could provide the shape of the input and output tensors for each convolutional layer in Fig. 9 or clarify in the text for better understanding.
Q2: U-Net is originally proposed for image segmentation, where the input is typically a 2D image. In your study, however, you applied U-Net to 1D signals. Could you clarify the specific modifications you made to adapt U-Net for processing 1D data?
Q3: The training details are insufficient. To enable reproduction of your work, please provide more comprehensive information, including the learning rate, optimizer, schedulers if there is any, and any other hyperparameters used during training.
Q4:Please provide additional details about your cross-validation procedure. Please include the number of folds in text.
Reviewer 3 Report
Comments and Suggestions for Authors
Review report on the ms entitled “Accurate Cardiac Duration Detection for Remote Blood Pressure
Estimation using mm-Wave Doppler Radar” by Shengze Wang, Mondher Bouazizi, Siyuan Yang, Tomoaki Ohtsuki.
The ms focuses on an improved methodology to accurately extract timing patterns of the heart activity by remote sensing methods, using Doppler radar technology. The topic is of general interest not only for the monitoring of cardiac patients, but also for the potential improvement and speed-up of express medical check-ups, either in general population or at workplaces where pre- and on-duty blood pressure control may be an essential part of requirements, such as operators of some critical infrastructure, emergency services, transportation etc.
The ms is generally well written, the key goals are clearly stated, there are clearly distinguishable novelty components, including the use of high-frequency oscillations, that were not taken into account in previous works. Therefore, the ms could be recommended for publication.
There are few aspects that could be addressed in the final version prior to publication:
1) The authors refer to the dataset they used for the validation as been recorded in supine position only. However, the study the authors are referring to under Ref. [17] considered not only the initial supine positions recordings, but also very interesting transitional regimes including the Valsalva manoeuvre (VM), holding breath, and the tilt table test, where pronounced variations in both heart rate and blood pressure are expected, and thus also heart function timing should be affected as well. Moreover, this study mentions 18 male and 7 female subjects, while the quoted study mentions 14 male and 16 female subjects.
Does this mean that in this paper only a subset of the quoted dataset was used? Was the data from some of the subjects not suitable for the analysis? Then it is important to understand what was the reason for that.
Moreover, was only the supine fraction of the data taken into account? Then why transitional regimes were excluded? Is there any inherent drawback preventing from their analysis using the proposed methodology? This would sound as a considerable limitation, and should be clarified explicitly.
Sorry if there is any misunderstanding on my side, but I believe these potential applicability issues and/or inherent limitations should be clarified.
2) Another limitation may be in the fact that the study focuses on healthy subjects with an average age around 30+/-10y. However, this is indeed not the most essential cohort for regular heart timing and blood pressure monitoring. In patients with long-term hypertension relation between cardiac timing and blood pressure is likely different than in healthy subjects due to arterial stiffness, impaired baroreflex and several other factors. This does not mean the proposed methodology will not be applicable in this scenario, but maybe it would require re-training or even simple calibration for a specific patient, e.g., at the time of admittance?
I am fully convinced that these rather technical issues could be addressed in a form of minor amendment.
